# Find2Find:
# Multitask Learning for Anaphora Resolution and Object Localization

**Cennet Oguz**[1]**, Pascal Denis**[2]**, Emmanuel Vincent**[3]
**Simon Ostermann**[1]**, and Josef van Genabith**[1]

[1]German Research Center for Artificial Intelligence (DFKI), Saarland Informatics
[2]Univ. Lille, Inria, CNRS, Centrale Lille, UMR 9189 CRIStAL, F-59000 Lille, France
[3]Université de Lorraine, CNRS, Inria, LORIA, F-54000 Nancy, France
{cennet.oguz, simon.ostermann, josef.van_genabith}@dfki.de
{pascal.denis, emmanuel.vincent}@inria.fr

## Abstract

In multimodal understanding tasks, visual and linguistic ambiguities can arise. Visual ambiguity can occur when visual objects require a model to ground a referring expression in a video without strong supervision, while linguistic ambiguity can occur from changes in entities in action flows. As an example from the cooking domain, "oil" mixed with "salt" and "pepper" could later be referred to as a "mixture". Without a clear visual-linguistic alignment, we cannot know which among several objects shown is referred to by the language expression "mixture", and without resolved antecedents, we cannot pinpoint what the mixture is. We define this chicken-and-egg problem as *visual-linguistic ambiguity*. In this paper, we present *Find2Find*, a joint anaphora resolution and object localization dataset targeting the problem of *visual-linguistic ambiguity*, consisting of 500 anaphora-annotated recipes with corresponding videos. We present experimental results of a novel end-to-end joint multitask learning framework for Find2Find that fuses visual and textual information and shows improvements both for anaphora resolution and object localization as compared to a strong single-task baseline.

## 1 Introduction

Deep neural networks have achieved enormous success in various language and computer vision tasks, such as multimodal understanding using video-text and image-text data (Malmaud et al., 2015; Alayrac et al., 2016; Zhou et al., 2018b; Miech et al., 2019; Zhukov et al., 2019). However, many current systems require a large number of accurate annotations, including image-level labels, location-level labels (bounding boxes and key points), and pixel-level labels.

A specific type of video data with naturally occurring semi-aligned texts are narrated instructional videos. Such videos are available in large quantities (e.g. on YouTube) and often chosen for learning

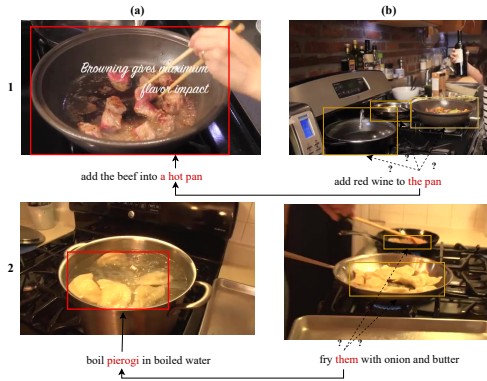

Figure 1: Examples of visual and linguistic ambiguities. Figure 1) represents the visual ambiguity related to which specific *pan* (in Figure 1a) is referenced with the phrase *the pan* because many pans occur on the stove. Figure 2) shows the linguistic ambiguity with the use of the pronoun *them* (in Figure 2b).

joint text-video embeddings in multimodal understanding (Zhou et al., 2018b; Miech et al., 2019). They often contain a narration explaining the visual content of the corresponding time frames in the video (Malmaud et al., 2015; Alayrac et al., 2016; Zhou et al., 2018b; Miech et al., 2019; Zhukov et al., 2019).

Instructional videos often contain visual and linguistic ambiguities that can be easily resolved by humans but pose two unique key challenges for automatic text-image processing systems. The first challenge is *visual ambiguity*, occurring when it is necessary to ground a referring expression in an image with ambiguous visual referents. In Figure 1 (1b), it is not clear from the picture which pan is referred to with the noun phrase *the pan* without a correct bounding box annotation. The second key challenge is *linguistic ambiguity*, instantiated for example by the use of anaphoric pronouns (Figure 1, (2b) or null pronouns (Figure 2).

In this work, we focus on modeling cases where both ambiguities are intertwined, a phenomenon we refer to as *visual-linguistic ambiguity*. Figure

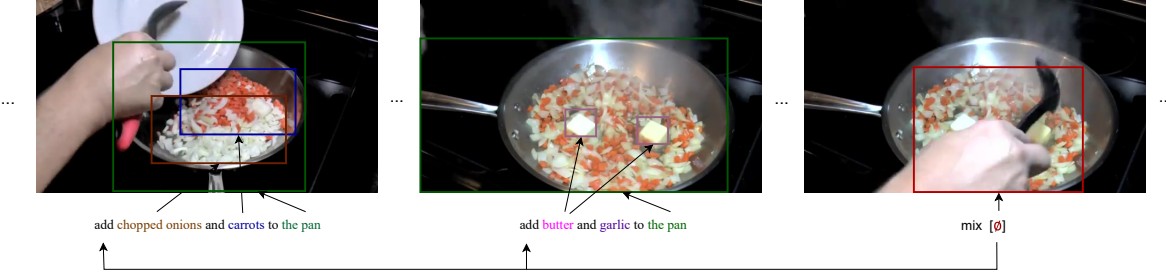

add chopped onions and carrots to the pan    add butter and garlic to the pan    mix [ø]

Figure 2: An example to display how visual-linguistic ambiguity occurs with a zero anaphor. The zero anaphor $[\phi]$ refers to two previous instructions as shown. The entities are aligned to the object with the arrows and the color codes.

1 provides a motivating example: To find which *pan* in (1b) is denoted by the textual span *the pan*, we need to find its antecedent in the preceding text. The visual object localization of the textual antecedent *the hot pan* in (1a) then includes supplementary visual information about the correct *pan*: The referent is *the hot pan* with beef in it. In Figure 1, To find the correct location of the object referred to as *them* in (2b), we first need to find the textual antecedent to understand what *them* refers to. When identifying *pierogi* as antecedent, we can use the *pierogi* for the visual object localization of *them*.

An even more complex case is null pronouns (Figure 2): To do object localization and anaphora resolution, the first requirement is the detection of the null pronoun, which then needs to be resolved and located in the image. A located visual object of a null pronoun then assists in finding the textual antecedent, and a resolved null pronoun helps to apply object localization.

Our guiding idea in this work is that *anaphora resolution*, the task of connecting linguistic expressions such as the anaphor (i.e., the repeated reference) and its antecedent (i.e., the previous mention in the document). (Poesio et al., 2018; Fang et al., 2022; Oguz et al., 2022), and *object localization*, the task of identifying the location of one or more objects in an image and drawing bounding boxes around their visual space (Tompson et al., 2015; Zhou et al., 2016; Choe et al., 2020), can jointly help to resolve visual-linguistic ambiguities. To test this we propose a multitask learning neural model for jointly resolving visual-linguistic ambiguity.

Our contributions are two-fold: **First**, we present a new dataset[1] (Section 4), *Find2Find*, for the joint evaluation of anaphora resolution and object localization based on an extension of *Chop&Change*

(Oguz et al., 2022). Our new data set contains 500 recipes with annotated anaphora and associated object localization. Together with the new data set, we propose the new task of multimodal resolution of *Visual-Linguistic Ambiguities*. The task provides a unique opportunity for models to fuse text and vision information for solving anaphora resolution and object localization at the same time. **Second**, we present a new multitask learning system[1] for modeling the two tasks of anaphora resolution and object localization jointly, using a fusion of visual and textual data. Our experiments show that information from each of the tasks mutually benefits performance on the other task. Our idea is based on the fact that in both tasks, the goal is to extract mentions from a given text: what connects object localization and anaphora resolution is that in visual object localization, a corresponding language expression needs to be found, and in anaphora resolution, accurate spans that resolve the anaphoric relations between the anaphor and the antecedents need to be identified.

## 2 Related Work

### 2.1 Anaphora Resolution

Anaphora resolution (Poesio et al., 2018; Fang et al., 2022; Oguz et al., 2022) is the process of resolving the relations between an anaphor (i.e., a reference expression) and its antecedent (i.e., the previous mention of the same entity). The relations between anaphor and antecedent mostly appear in two different forms: *coreference* and *bridging*. Coreference resolution (Clark and Manning, 2016a; Lee et al., 2017) is the task of finding the linguistic expressions that refer to the same real-world entities in a document, whereas bridging resolution (Yu and Poesio, 2020; Kobayashi et al., 2022) fo-

---

[1]https://github.com/OguzCennet/odar

cuses on entities with an associative relation that does not express the same entity but relates to it (e.g., a car and its engine). Most previous work and datasets (Yu et al., 2022) tackle coreference resolution and bridging resolution separately. An exception is Fang et al. (2021), Fang et al. (2022), and Oguz et al. (2022), which focus on documents with rich anaphoric relations for anaphora annotation and resolution of coreference *and* bridging with end-to-end neural networks (Lee et al., 2017).

Anaphora resolution is composed of two subtasks: *mention detection* and *antecedent selection*. A typical neural-based method for anaphora resolution starts initially with neural-based mention-ranking modeling (Clark and Manning, 2016a), using distributional features of entities (Clark and Manning, 2016b) with predefined mentions. Lee et al. (2017) combine mention detection and antecedent selection in an end-to-end neural learning system. Yu and Poesio (2020) propose a multi-task learning system for coreference and bridging resolution with an end-to-end learning approach (Lee et al., 2017). Here, coreference and bridging resolution models learn the mention detection with the objective of coreference/bridging resolution: if the span is resolved then it is a mention.

Various feature sets are used for anaphora resolution along with contextualized language features (Joshi et al., 2019, 2020), e.g., the token length of spans (Clark and Manning, 2016b), context-dependent boundary representations with a head-finding attention mechanism over the span (Lee et al., 2017), distance features of the anaphor and the antecedent based on word distance (Clark and Manning, 2016a) and sentence distance (Oguz et al., 2022) where additionally visual features are used for anaphora resolution in recipes for cooking videos. Our work is similar in spirit: however, unlike Oguz et al. (2022) we combine anaphora resolution with object localization tasks from computer vision in a joint multitask learning model to benefit both leveraging visual features of entities for anaphora resolution.

## 2.2 Object Localization

Object localization is the task of identifying the location of one or more objects in an image, e.g. Figure 3 (a,b,c). Object localization has been studied in computer vision for a long time with various learning methods (Gokberk Cinbis et al., 2014; Zhou et al., 2018a; Huang et al., 2018).

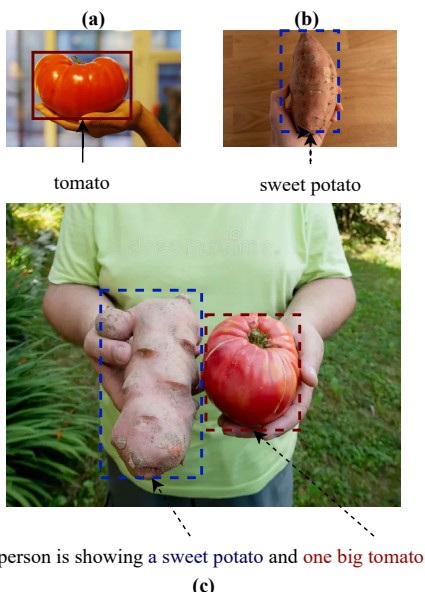

a person is showing a sweet potato and one big tomato
**(c)**

Figure 3: Different examples of object localization. The dashed lines in (b) and (c) represent the occurrence of annotation for only the test data whereas the straight lines (a) indicate that annotation is present for train and test.

**Multiple Instance Learning (MIL).** MIL (Dietterich et al., 1997) is a learning strategy that adresses the essence of the incomplete annotation problem, where only coarse-grained labels are available for learning. MIL has been used effectively for weakly-supervised learning in several computer vision works including object tracking (Babenko et al., 2010), object localization (Gokberk Cinbis et al., 2014), image classification (Wu et al., 2015). Huang et al. (2018) extend MIL reference awareness for visual grounding of instructional videos. We extend MIL for object localization with anaphoric information to avoid the issue of ambiguous language references such as *it*, *them* (Zhou et al., 2018a; Huang et al., 2018).

**Weakly Supervised Object Localization** The most common way to do object localization is supervised learning, which uses object-level categories with bounding boxes, e.g., Figure 3 a. However, for a data-greedy neural learning system, object-level bounding box annotation is time-consuming and expensive. Weakly supervised object localization approaches avoid this problem and focus on learning object localization with image-level object labels (e.g., Figure 3 b) under the MIL paradigm (Deselaers et al., 2012; Prest et al., 2012; Gokberk Cinbis et al., 2014; Oquab et al., 2015).

Object localization with the MIL approach aims to match object labels with object bounding boxes (e.g., Figure 3 b). Studies on object localization with image descriptions rather than object labels take weak supervision further (Karpathy and Fei-Fei, 2015; Zhou et al., 2018a; Huang et al., 2018), as in Figure 3 c. For example, Huang et al. (2018) propose to extend object localization to localization based on context-dependent referring expressions, and (Kuo et al., 2022) offers a general-purpose model for object localization by using a wide range of referring expressions, localization or detection queries for zero, one, or multiple objects. Another challenge for object localization with image descriptions is the automatic extraction of object labels from image descriptions. Zhou et al. (2018a) extract object labels manually for training and testing object localization models whereas Huang et al. (2018) apply the pre-trained Stanford CoreNLP parser (Manning et al., 2014) for entity detection.

To date, object localization studies have not explored learning from to learn of mention extraction from image descriptions, instead extracting the mentions by using parsing methods (Kiddon et al., 2015; Huang et al., 2017, 2018) or using the predefined mentions list (Zhou et al., 2018a) before the learning process. Thus, we claim object localization and anaphora resolution share a subtask of entity extraction like the mention detection process in anaphora resolution as explained in Section 2.1

## 3 Task

A recipe consists of instructions $I$ where a cooking instruction $I_i$ (e.g. *add chopped onions and carrots to the pan*) consists of $n$ nominal or null spans and one verbal predicate where $n \geq 1$. A span $x_i$ of $I_i$ might be an incorrect consecutive fragment *add chopped* or a gold span $e$, e.g., a noun phrase *chopped onions*, a pronoun *it*, or a null pronoun $[\phi]$ as in *mix* $[\phi]$. Null pronouns are extremely common in recipe instructions (Kiddon et al., 2015; Huang et al., 2017). In our approach, we also have a video clip that contains the visual content of the instruction $I_i$ with the action and the entities included in the process. Following Zhou et al. (2018a); Huang et al. (2018), we evenly divide each video clip into three equal parts and randomly sample one image (one frame) $V_i$ from each of the three sub-clips to capture the temporal changes of entities.

| | Train | Test |
|---|---|---|
| Entity | 9,316 | 2,842 |
| Null Pronoun | 1,002 | 282 |
| Pronoun | 314 | 129 |
| Noun Phrases | 8,000 | 2,431 |
| Instruction | 4,633 | 1,422 |
| Recipe | 400 | 100 |

Table 1: Annotated Data Statistics

### 3.1 Anaphora Resolution

The task of anaphora resolution is to assign each gold span (anaphor) $e_i$ where $e_i \in \{x_{i,1}, \ldots, x_{i,1}\}$ of each instruction $I_i$ to one or more gold spans (antecedent) $y_i \in \{\epsilon, I_1, \ldots, I_{i-1}, e_{1,1}, \ldots, e_{i-1,n}\}$, a dummy antecedent $\epsilon$, all preceding instructions $I_1, \ldots, I_{i-1}$ and all preceding gold spans $e_{1,1}, \ldots, e_{i-1,n}$ from the previous instructions $I_1, \ldots, I_{i-1}$. For a nominal span in Figure 1 1b, the anaphor span *the pan* refers to the antecedent *a hot pan* in a previous instruction. For a null pronouns example in Figure 2, the null pronouns $\phi$ refers to two previous instructions as the antecedents because the null pronoun does not point to any entity and it is also not a new entity for the recipe, it is instead produced by the previous instructions.

The selection of dummy $\epsilon$ as antecedent indicates that the anaphor is an incorrect sequence of consecutive words or a singular entity without an antecedent.

### 3.2 Object Localization

An image $V \in \mathbb{R}^{H \times W}$ is represented as a bag of ten regions (e.g., the visualization of region proposals of the given positive frame in Figure 4), $v_1^{h \times w}, .., v_{10}^{h \times w}$ with suitable side lengths of window patches $h$ and $w$ where $h < H$ and $w < W$. Given a span $x_i$ and ten region proposals $v_1^{h \times w}, .., v_{10}^{h \times w}$, the task of object localization is to identify whether or not the region proposal belongs to the object of interest, i.e., the text $x_i$ when $x_i$ is a gold span (e.g., null pronoun $[\phi]$ or nominal spans) of the corresponding instruction $I_i$. Therefore, the task is to link the text span $x_i$ to a corresponding visual region proposal $v_i$ where $x_i$ is a gold span $\{e_{i,1}, \ldots, e_{i,n}\}$ of the instruction. We use the dot-product attention between the span $x_i$ and the region $v_i$ for ranking the visual-semantic matching.

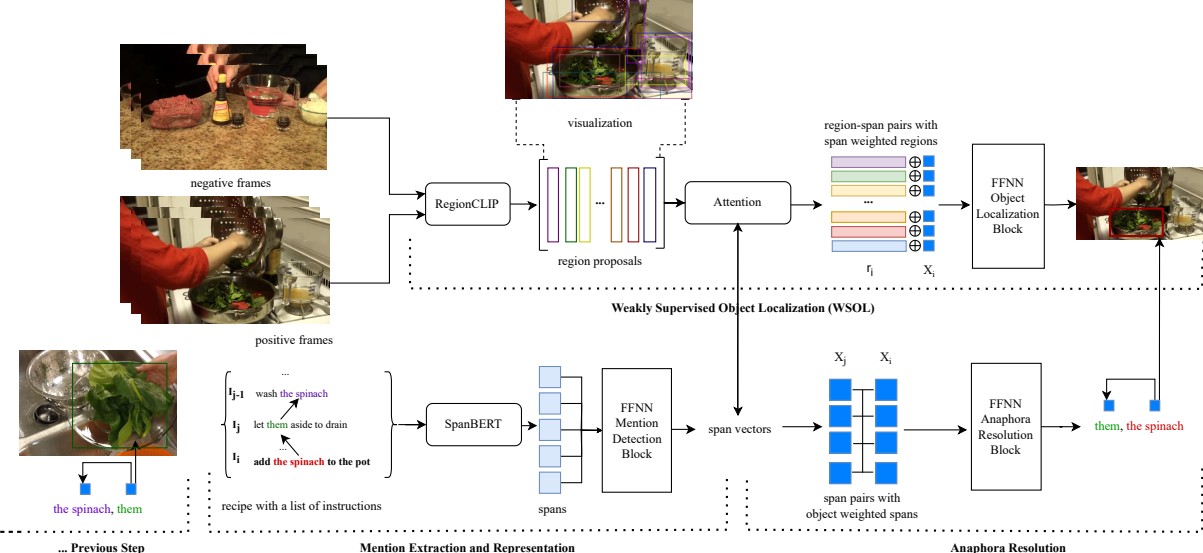

Figure 4: The architecture of the multitask learning framework of anaphora resolution and object localization.

## 4 Data

**Anaphora Resolution Data.** The YouCookII dataset (Zhou et al., 2018a,b) includes manually provided descriptions (i.e., instructions) of actions with the corresponding temporal boundaries (i.e. start and end timestamps) in 2,000 cooking videos. The videos provide visual input of the corresponding objects to observe changes of the objects clearly. Oguz et al. (2022) use the YouCookII dataset to propose a multimodal anaphora resolution dataset, Chop&Change, with a novel annotation schema to address the state change of entities in cooking instructions of recipe videos. While the Chop&Change annotation schema captures three anaphoric relations (coreference, near-identity, and bridging), in our work here we concentrate on anaphora resolution ignoring relations and focusing on finding the antecedent. Table 1 shows that the Chop&Change dataset includes 264 training recipe documents and 89 test documents in total. For our work here, we increase the number of annotated recipes to 400 for train and 100 for test recipes by using the Chop&Change annotation schema. All annotated recipes are associated with respective videos. The structure of annotation is explained with an example in Appendix A.

**Object Localization Data.** To construct our test set we examine Huang et al. (2018) who present a study on reference-aware visual grounding and provide an object localization dataset, *FindIt*, of 62 YouCookII videos for the given entities in the pro-

vided textual descriptions (i.e., instructions). After a deep examination of the FindIt dataset, we find that 30 videos of FindIt could be used with our annotated recipes. Since only 30 videos produce an insufficient test set for the evaluation of object localization, we annotated 70 more videos from the recipes we annotated for anaphora resolution. We extract the frames of the instruction clips by using the temporal boundaries. We obtain 3 video subsets of consecutive frames of the instruction to acquire the state changes of entities in time. We pick one frame with a clear visual content of each entity from the 3 subsets. Thus, each entity is represented in a maximum of three frames for state changes. For evaluation, we annotate the visual object with a bounding box on the selected frames for each entity. In total, we have **5,688 images for 100 recipes** annotated for our object localization test set. Note that we do not annotate bounding boxes for the training data as we train our models in a weakly supervised setting.

## 5 Methodology

### 5.1 Model

In this section, we explain the details of our anaphora resolution and object localization models. Additionally, we formulate the multitask learning approach of joint anaphora resolution and object localization (see Figure 4). In order to analyze a given text, it is important to resolve referring expressions. Thus, text-based anaphora resolution is an important method of identifying the antecedent

of anaphora. However, here we also have object localization, providing a bounding box for the visual content of language references. The tasks of object localization and anaphora resolution share information via the mention extraction and representation part of the model. Therefore, our mention representations are trained with anaphora resolution and object localization over the mention extraction and representation. The implementation details can be seen in Appendix B.

As implied in the explanation of both tasks, before resolving references in the previous context or on the image, referring expressions and language references in the text need to be identified, a task known as *Mention Extraction*. In the next Section, we first describe our approach to representing mentions.

### 5.1.1 Mention Extraction and Representation

We consider all continuous token sequences with up to $L$ words as a potential mention span and compute the corresponding span score. We use SpanBERT (Joshi et al., 2020) as a state-of-art representation for coreference resolution. We capture the linguistic dependencies between anaphor and antecedent in a recipe document by exploiting self-attention: we define $\text{SPANBERT}(w_1, \ldots, w_T)$ to be the SpanBERT representation of a recipe, where $w_1$ is the first token and $w_T$ refers to the last token of the recipe. SpanBERT captures the long-range dependencies between the antecedent and anaphor in the recipes. A span $x_i$ consists of zero or more tokens of instruction $I_i$. We use the verb as a pointer for null pronouns. For example, *mix* is the token of the null pronoun $\phi$ in the instruction *mix* $[\phi]$ (Figure 2). The vector representation $g_i$ of a given span $x_i$ is obtained by concatenating the contextualized SpanBERT word vectors of its boundary tokens and its width feature:

$$g_i = [x^*_{\text{START}(i)}, x^*_{\text{END}(i)}, \phi(i)]$$
$$\phi(i) = \text{WIDTH}(\text{END}(i) - \text{START}(i)).$$

$\text{START}(i)$ and $\text{END}(i)$ represent the starting and ending token indexes for $g_i$, respectively. $\phi(i)$ is the width feature of the span where $\text{WIDTH}(.)$ is the embedding function of the predefined bins of $[1, 2, 3, 4, 8, 16]$ as defined by Clark and Manning (2016b).

### 5.1.2 Weakly Supervised Object Localization

Following prior work (Karpathy and Fei-Fei, 2015; Huang et al., 2017, 2018), we observe that sentence descriptions of pictures make frequent references to objects in the pictures and their attributes. However, the references are not always clearly defined. Particularly in a video, the use of pronouns and ellipses are extremely common (Kiddon et al., 2015; Huang et al., 2017). Our object localization model follows a Weakly Supervised Object Localization (WSOL) strategy (Huang et al., 2018; Zhou et al., 2018a; Choe et al., 2020): Only full image descriptions (Figure 3 (c)) are used for localization instead of a specific object-level label (as in bounding box and specific label pairs for each object in a picture like in Figure 3 (a)).

Following the object region and text ranking approach, we formulate the task of WSOL as mapping the frame region $v_i$ to the text span $x_i$. We use positive and negative frames where positive frames come from the video clip $V_i$ of instruction $I_i$ whereas negative frames are drawn from other videos without shared entities. We define $\text{pos}_i$ to be a region vector set that includes the region proposals $v_1^{h \times w}, .., v_{10}^{h \times w}$ from the *positive* image $V_i$, and $\text{neg}_i$ to contain the region proposals $v_1^{h \times w}, .., v_{10}^{h \times w}$ from the *negative* image, e.g., the negative and positive frames in Figure 4.

The aim of WSOL is to produce a scoring function to maximize the joint probability of positive frame regions $v_i \in \text{pos}_i$ and minimize the joint probability of negative regions $r_i \in \text{neg}_i$ with the text span $x_i$. We concatenate the mention representation vector $g_i$ of span $x_i$ and the visual object region vector $r_i$ of the region $v_i$ to obtain the WSOL input, $[g_i, r_i]$, effectively fusing textual and visual information in one input. We then prepare positive ($\text{FFNN}(g_i, r_i) = 1$) and negative ($\text{FFNN}(g_i, r_i) = 0$) samples to train a model with attention-based deep MIL (Ilse et al., 2018). Positive examples depict exactly the action in question, whereas negative examples correspond to one of four special cases as described below:

$$\text{FFNN}(g_i, r_i) = \begin{cases} 0 & x_i = \epsilon, \ \forall r_i \\ 0 & x_i \notin \{e_{i,1}, \ldots, e_{i,n}\}, \ \forall r_i \\ 0 & x_i \in \{e_{i,1}, \ldots, e_{i,n}\}, \ r_i \in \text{neg}_i \\ 1 & x_i \in \{e_{i,1}, \ldots, e_{i,n}\}, \ r_i \in \text{pos}_i \end{cases}$$

Our localization model uses the span representation $g_i$ that is extracted by the mention extraction and ten positive, i.e., $\text{pos}_i$, and ten negatives, i.e., $\text{neg}_i$, region representation vectors $r_i$ to learn the best region from $\text{pos}_i$ for the given span $g_i$. Thus, our mention detection model learns the span vector $g_i$

also based on the object localization objective (see also Figure 4). We define the label of FFNN($g_i, r_i$) as 1 when the span $x_i$ is a gold span $\{e_{i,1}, \ldots, e_{i,n}\}$ and the region vector $r_i$ represent the positive regions pos$_i$.

### 5.1.3 Anaphora Resolution

Following Lee et al. (2017) and Oguz et al. (2022), we implement our anaphora resolution system as an end-to-end system with mention detection but now extended by object localization. For anaphora resolution, the representation of a span pair $g_{ij}$ is obtained by concatenating the two span embeddings $[g_i, g_j]$ and their element-wise multiplication, $g_i \cdot g_j$, among others:

$$g_{ij} = [g_i, g_j, g_i \cdot g_j, \phi_{dist}(i,j)]$$

where the feature vector $\phi_{dist}(i,j)$ is the distance DISTANCE(START($j$) − START($i$)) between the index of the instruction span $i$ and span $j$. DISTANCE(·) is an embedding function of the predefined bins of $[1, 2, 3.., 30]$ as in Oguz et al. (2022). We use softmax(FFNN($g_{ij}$)) to score the resolution for anaphor $g_i$ and antecedent $g_j$ pairs.

### 5.2 Evaluation

Following Hou et al. (2018) and Yu and Poesio (2020), we assess the performance of our end-to-end anaphora resolution with the F1-score where precision is the result of dividing the number of correctly predicted pairs by the total number of predicted pairs and recall is computed by dividing the number of correctly predicted pairs by the total number of gold pairs.

To evaluate object localization, we follow prior work (Fukui et al., 2016; Rohrbach et al., 2016; Huang et al., 2018) and compute accuracy as the ratio of phrases for which the predicted bounding box overlaps with the ground-truth by more than 0.5 Intersection-over-Union (IoU).

## 6 Experimental Setup

### 6.1 Input

#### 6.1.1 Cooking Instructions.

To encode the recipes we use spanBERT (Joshi et al., 2020), a transformer model designed to better represent and predict spans of text. We use the concatenation of the boundary tokens to represent each span (Clark and Manning, 2016a,b; Lee et al., 2017; Kobayashi et al., 2022).

#### 6.1.2 Frame Regions of Instructions

Image regions can be proposed by either off-the-shelf object localizers, e.g., region proposal networks (Girshick et al., 2014; Ren et al., 2015), or dense sliding windows (e.g., random regions). Region proposal networks and dense sliding window methods neglect language semantics. Therefore, we leverage the state-of-art method *RegionCLIP* (Zhong et al., 2022) which uses a *CLIP* model (Radford et al., 2021) to match image regions with template captions to align these region-text pairs in the feature space. We select the first 20 region proposals of each entity with the highest objectness scores.

### 6.2 Experiments

#### 6.2.1 Anaphora Resolution

**Candidate and Gold Spans.** Without any pruning, we consider all continuous token sequences (Clark and Manning, 2016b; Lee et al., 2017) as potential spans for anaphor/antecedent candidates for the training and testing phases. Additionally, we consider gold spans for the training and testing phases in order to investigate the performance of anaphora resolution models without mention detection noise.

**With and without Object Localization.** To understand the effect of object localization on anaphora resolution we examine the anaphora resolution results with and without object localization. Here, we remove the object localization model and data from the training and testing phases of our anaphora resolution model.

#### 6.2.2 Object Localization

**With and without Anaphora Resolution.** We investigate the object localization performance with and without anaphora resolution to understand the impact of anaphora resolution. We perform four different experiments for object localization.

**Random** (Huang et al., 2018): Since we use the object region proposal with the highest objectness scores with the given ten region proposals, we first examine random selection with the highest objectness score to show the complexity of object localization.

**Deep Visual-Semantic Alignment (DVSA)** (Karpathy and Fei-Fei, 2015): This weakly supervised visual grounding method is based on image-based regions and given gold and candidate

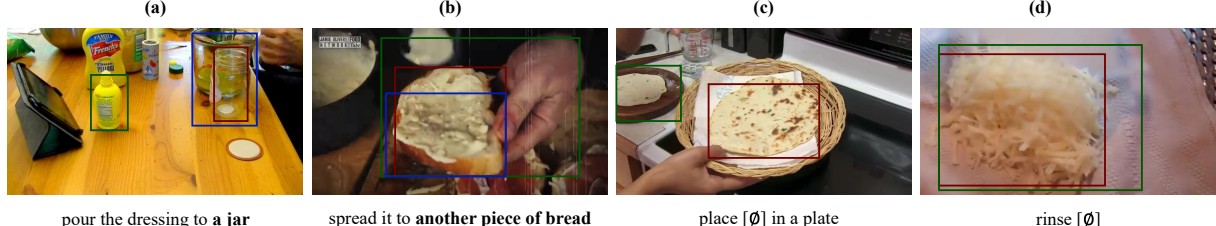

| (a) | (b) | (c) | (d) |
| --- | --- | --- | --- |
| pour the dressing to **a jar** | spread it to **another piece of bread** | place [∅] in a plate | rinse [∅] |

Figure 5: The examples of object localization results of *random* with the green bounding box, *DVSA* with the blue bounding box, and our multitask learning method with the red bounding box. The task is the object localization of the gold mention in bold font in the instructions.

| Methods | Nominal | Null | All |
| --- | --- | --- | --- |
| Random | 13.98 | 16.66 | 14.07 |
| DVSA w Gold Mentions | 19.90 | - | 19.90 |
| AR w Cand. Mentions | **21.02** | **24.46** | **20.79** |
| AR w Gold Mentions | **21.17** | **25.98** | **22.36** |

Table 2: The Top-1 results of object localization with gold and candidate mentions. The column group named Nominal shows the results of object localization for nominal phrases, Null depicts the results of null pronouns, and All refers to the full dataset.

mentions without anaphoric information, which uses multiple-instance learning. Thus, we consider DVSA as a baseline for weakly supervised object localization to our method without anaphora resolution. We examine the results of DVSA with gold and candidate mentions.

**AR-Cand. Mentions** We train and test the object localization model with all continuous tokens (Clark and Manning, 2016b; Lee et al., 2017) as potential spans for anaphor/antecedent candidates. Candidate spans also increase the noise in the object label set because we consider all possible spans as labels of the objects.

**AR-Gold Mentions** We show the results of object localization when we use gold mentions and anaphor information for training and testing.

# 7 Results and Discussion

## 7.1 Overview

We investigate the anaphora resolution and object localization results of gold and candidate spans comparing the F1-scores and Top-1 scores with multitask learning using both tasks and single tasks. Overall, our results in Table 3 demonstrate that replacing single-task learning with our multitask joint learning approach improves anaphora reso-

lution and object localization for both candidate and gold spans. The difference between the results of candidate and gold spans demonstrates that the mention extraction model propagates errors to anaphora resolution and object localization (the sequential structure of model see Figure 4). For example, we have the candidate spans such as all n-grams words (bigrams words such as *cook the*, *the bacon*, *bacon fat*) of an instruction *cook the bacon fat* with the same visual features with bacon and fat in a pan. Thus, an incorrect mention of mention detection directly causes an error in object localization and anaphora resolution.

## 7.2 Weakly Supervised Object Localization

Overall, we observe an improvement in the performance of object localization with multitask learning shown in Table 2. Table 2 shows that our joint multitask learning method outperforms DVSA even with candidate mention. Note that DVSA is not the component that is responsible for extracting the mentions for object localization. Thus, we do not analyze object localization for null pronouns and candidate mentions because DVSA ranks the language expression and region pairs and does not apply mention extraction. However, we attached the DVSA method for ranking the gold nominal mentions with the given region proposals.

The results of object localization with null pronouns clearly demonstrate the benefits of anaphora resolution for object localization. For example, our model localizes the singular mention *a jar* in Figure 5 (a), and the anaphor *another piece of bread* in Figure 5 (b) better than other methods. Thanks to the mention detection and representation of our multitask learning approach, our object localization model is capable of localization of null pronouns, e.g., Figure 5 (c,d) as our model learns to represent the null pronouns in the mention detection process as a mention. Thus, the results of null pronouns

| | Nominal Anaphora Res. | | | Zero Anaphora Res. | | | Anaphora Res. | | |
|---|---|---|---|---|---|---|---|---|---|
| **Methods** | Precision | Recall | F1-score | Precision | Recall | F1-score | Precision | Recall | F1-score |
| **w/o Object Loc.** | | | | | | | | | |
| Cand. Mentions | 54.76 | 46.65 | 50.38 | 73.38 | 68.76 | 71.00 | 63.03 | 54.06 | 58.20 |
| Gold Mentions | 58.76 | 52.25 | 55.31 | 75.38 | 71.18 | 73.22 | 64.16 | 58.15 | 61.01 |
| **w Object Loc.** | | | | | | | | | |
| Cand. Mentions | 52.03 | 50.49 | 51.25 | 77.68 | 69.97 | 73.63 | 62.46 | 56.19 | 59.16 |
| Gold Mentions | 58.24 | 55.43 | **56.80** | 80.10 | 76.02 | **78.01** | 64.92 | 61.93 | **63.39** |

Table 3: Results of the anaphora resolution with and without object localization for gold and candidate mentions. We show the results for the full test datasets in Anaphora Res. columns, for only null pronouns in Zero Anaphora Res., and the resolution results of anaphoric mentions for all nominal phrases in Nominal Anaphora Res. part.

clearly evidence the contribution of anaphora resolution for object localization.

## 7.3 Anaphora Resolution

Multitask learning of anaphora resolution and object localization increases the performance of anaphora resolution. Table 3 shows $> 2\%$ improvements for anaphora resolution with gold mentions and more than $1\%$ for candidate mentions with combined training and testing. When nominal and zero anaphora are investigated separately, the results of nominal anaphora demonstrate an improvement on the results of gold mentions when object localization is included. Additionally, the results of zero anaphora show significant improvements with multitask learning for gold and candidate mentions. Thus, we observe a big part of the improvement for the combined experiments with object localization comes from the zero anaphora resolution samples. The benefit of object localization on anaphora resolution is also seen in candidate nominal mentions; it is however not as significant as in gold nominal mentions, as the errors of mention detection for candidate spans directly cause localization as well as anaphora resolution errors.

The most common error in anaphora resolution is to find the closest antecedent in the entity chain. For example, for *tomato→ it → tomato*, the first *tomato* is the antecedent of *it*, and *it* is the antecedent of the second *tomato*. However, the model fails to select the first *tomato* as the antecedent for the second one.

## 8 Conclusion and Future Work

In this work, we study the problem of *visual-linguistic ambiguity* in multimodal data and propose the novel task of joint anaphora resolution and object localization. We create the Find2Find dataset for object localization and anaphora resolution with cooking recipes to improve the performance of both tasks by exploiting their commonalities. We implement a model for the joint learning of anaphora resolution and object localization, fusing visual and textual information, and show empirically that a multitask learning paradigm mutually improves both tasks. Especially, the results of zero anaphora resolution indicate that object localization helps to avoid linguistic ambiguity of null pronouns. Our joint multitask learning approach does not apply the temporal features of evolving visual entities even though the temporal features of textual recipes (Oguz et al., 2022) are included due to instruction order. In future work, we claim the temporal encoding of visual objects can improve the results of joint anaphora resolution and object localization.

## 9 Acknowledgements

We would like to thank Gokul Srinivasagan and David Meier for helping with the annotation of the object localization, Alina Leippert for helping with the annotation of the anaphora resolution. This research was funded by the joint IMPRESS (01|S20076) project between the French National Institute for Research in Digital Science and Technology (Inria) and the German Research Center for Artificial Intelligence (DFKI).

## Limitations

The first and most important limitation is the pre-trained region proposal network (RegionCLIP (Zhong et al., 2022)). Our object localization is highly dependent on the quality of the region proposals. Therefore, a better region proposal network delivers more improvements for our multitask learning.

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

refers to *the onions* of the third instruction: *the onions → the onion → the onions*. Most of the errors of anaphora resolution for recipes occur when the model predicts *the onions* of the third instruction as the antecedent of the anaphor *the onions* of the sixth instruction. Additionally, Figure 6 shows the null pronouns in the eighth instruction and the antecedents such as the fifth, sixth, and seventh instructions.

## B    Implementation Details

We use Adam (Kingma and Ba, 2014) for optimization and a learning rate of 0.001. We clip gradients element-wise at 5 and use 0.3 dropouts for regularization. We use Negative log-likelihood as a loss function for the both task of anaphora resolution and object localization.

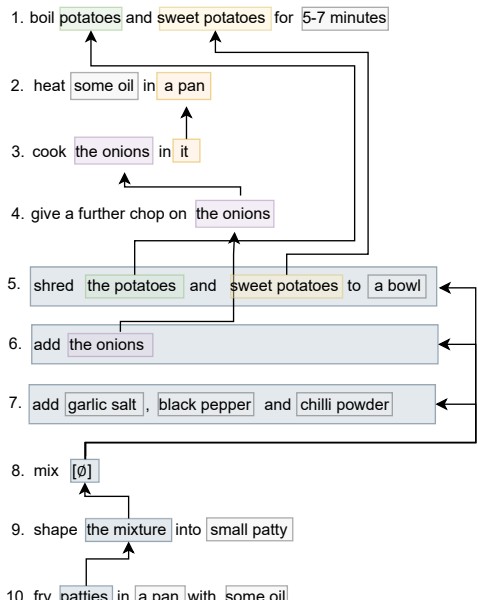

Figure 6: An example of the annotation of anaphora resolution for cooking instructions. The grey boxes represent the singular mentions without any anaphoric relations. The beginning of the arrows shows the anaphora whereas the ends point to the antecedent.

## A    Anaphora Annotation

Anaphora resolution is a challenging task for instructional languages because of temporally evolving entities (Oguz et al., 2022). For example, Figure 6 demonstrates how *the onions* of the sixth instruction step refers to *the onions* of the fourth instruction and *the onions* of the fourth instruction