# OpenReview forum: "Find-2-Find: Multitask Learning for Anaphora Resolution and Object Localization"
_EMNLP/2023/Conference — EMNLP 2023 Main_

### Official Review · Reviewer_pwhj · 2023-07-28

**Typos Grammar Style And Presentation Improvements:** 1. Clarity and Explanation of Symbols
**Soundness:** 4

**Excitement:**

4: Strong: This paper deepens the understanding of some phenomenon or lowers the barriers to an existing research direction.

**Paper Topic And Main Contributions:**

Paper Topic:

This paper addresses the challenge of visual-linguistic ambiguity in multimodal understanding tasks. Specifically, it tackles the issues arising from visual objects requiring the model to ground a referring expression in a video without strong supervision (visual ambiguity), and ambiguities that occur from changes in entities in action flows (linguistic ambiguity). To explore and resolve this chicken-and-egg problem, the authors propose a novel task of joint anaphora resolution and object localization.

Main Contributions:

1. Dataset Creation: The authors contribute a novel dataset, "Find2Find", designed specifically for the task of joint anaphora resolution and object localization. This dataset includes 500 anaphora-annotated recipes with corresponding videos, offering a rich resource for the study of visual-linguistic ambiguity.

2. Model Implementation: The paper introduces an innovative end-to-end multitask learning framework for handling the aforementioned dataset. This model is capable of jointly learning anaphora resolution and object localization, thereby fusing visual and textual information to address the problem of visual-linguistic ambiguity.

3. Empirical Validation: The authors provide experimental results demonstrating the effectiveness of their approach. The results reveal that the multitask learning paradigm mutually improves both anaphora resolution and object localization tasks. Moreover, the study shows that object localization can assist in avoiding linguistic ambiguity of null pronouns, thereby underlining the efficacy of the proposed method.

In sum, this paper contributes a new data resource and a practical approach to address the challenging problem of visual-linguistic ambiguity, offering advancements for the field of multimodal understanding tasks.

**Questions For The Authors:**

It is meaningful and exciting work with solid experiments. Here are my questions:
A. Can you improve the writing to prevent confusion reading the paper specified in the Typos Grammar Style And Presentation Improvements section?
B. Can you elaborate more on the data annotation details? Is there any quality check process?
C. Can you elaborate more on how this novel dataset and task can potentially help improve the visual-textual understanding abilities of other multi-modal domains?

**Reasons To Accept:**

1. Novel Dataset: The creation of the "Find2Find" dataset is a significant contribution. It addresses a gap in resources for studying visual-linguistic ambiguity, and provides a valuable resource for future research in this area.
2. Innovative Approach: The paper proposes a novel joint task of anaphora resolution and object localization. This new perspective and approach can inspire and inform subsequent research in multimodal understanding tasks.
3. Empirical Evidence: The authors provide empirical evidence to support their approach. This validation strengthens the credibility of the proposed method and demonstrates its practical applicability.
4. Benchmark: The proposed model and the empirical results can serve as a benchmark for future studies aiming to solve similar problems in the realm of multimodal understanding tasks.


**Reasons To Reject:**

I do not have much reason to reject it, but just a little concerned about its generalization ability. The "Find2Find" dataset and the proposed model are focused on the cooking domain. The effectiveness of the proposed method in other domains or more diverse contexts remains unexplored. The results and insights, thus, may not be generalizable.


**Reproducibility:**

4: Could mostly reproduce the results, but there may be some variation because of sample variance or minor variations in their interpretation of the protocol or method.

**Reviewer Confidence:**

4: Quite sure. I tried to check the important points carefully. It's unlikely, though conceivable, that I missed something that should affect my ratings.

---

> ### Author Rebuttal · Authors · 2023-08-28
>
> ### Questions A:
>
> > Many thanks for your careful and comprehensive feedback. We will carefully revise the writing of the paper to prevent possible confusion on the part of readers.
>
> ### Questions B:
> We have two different annotations in this study:
>
> #### Annotation for anaphora resolution:
>
> We follow the annotation guidelines of  Oguz et al. (2022) and show an example of anaphora annotation in Appendix A Figure 6.  We selected at least five videos for training and at least 1 video for testing each of the 89 different recipes. This helps us to increase the object variations of visual and language data in our experiments.
>
> 1. Boil **potatoes** and **sweet potatoes** for **5-7 minutes**
> 2. Heat **some oil** in **a pan**
> 3. Cook **onions** in **it**
> 4. Give a further chop to **onions**
> 5. Shred **the potatoes** and **the sweet potatoes** in **a bowl**
> 6. Add **the onion pieces**
> 7. …
>
> The bold entities are our mentions. A mention can be either an anaphoric mention or a singular mention (i.e., a common technical term for the mentions without anaphoric relation).  For example, the spans **some oil**, **5-7 minutes**, and **a bowl**  are singular mentions, The other mentions are anaphoric mentions. The mention ‘it’ in step 3 refers to **a pan** in step 2; the mention **onions** in step 4 refers to  **onions** in step 3; the mention **potatoes** in step 5 refers to  **the potatoes** in step 1,  the mention **the sweet potatoes** in step 5 refers to  **sweet potatoes** in step 1;   The mention **the onion pieces** in step 6 refers to **onions** in step 4.
>
>
> We do not apply any clustering of the mentions. For example the mentions **the onion pieces** in step 6, **onions** in step 4, and **onions** in step 3 do not get in one cluster. The mention of **the onion pieces** in step 6 refers to **onions** in step 4, and the **onions** in step 4 refers to **onions** in step 3. This is because the onions we see in step 6 are the onions chopped in step 4 and the onions in step 4 are the onions cooked in step 3.
>
> #### Annotation for object localization:
>
> Each mention is connected to the object bounding boxes of images with mention IDs.  For example, the ID of mention ‘it’ in step 3 is E10, the ID of the bounding box of the corresponding image is also E10. We obtain 3 video subsets of consecutive frames of the instruction to acquire the state changes of entities in time. We pick one frame with a clear visual content of each entity from the 3 subsets. Thus, each entity is represented in a maximum of three frames for state changes. Then we annotate each frame of an object with a bounding box around the object.
>
> ### Questions C:
>
> All instructional videos share a universal property, and that is that substances and objects in them undergo change, including chemical, physical, part-whole, and other changes. This makes such data extremely challenging for NLP, as both textual reference to objects and the objects themselves continuously change during the temporal progression of the video. What we present in this paper with cooking videos is highly relevant for industrial process videos, construction video, videos of scientific experiments, etc, and at the cutting edge of multimodal NPL towards mining such multimodal data. To date, there is very little research targeting this challenging setting and we see our research as an important contribution to advancing the field.
>
> An instruction example for the woodworking domain
> 1. put 3 equal size planks on the floor
> 2. screw them together
> 3. cut 4 pieces of wood to equal size
> 4. glue the legs to the tabletop
> 5. paint the table
>
> + 3 equal size planks, the floor are sungular mentions (no antecedent)
> + them in step 2 refers  3 equal size planks in step 1
> + the legs in step 4 refers  4 pieces of wood  in step 3
> + the tabletop in step 4 refers screw them together step 1
> + the table in step 5 refers glue the legs to the tabletop  step 4
>
> Let’s assume that we aim to find the object location of ‘them’ in step 2. We first need to resolve ‘them’ to find which object we should localize for the given image.

---

### Official Review · Reviewer_b6CE · 2023-08-05

**Soundness:** 4

**Excitement:**

4: Strong: This paper deepens the understanding of some phenomenon or lowers the barriers to an existing research direction.

**Paper Topic And Main Contributions:**

The paper proposes a new task of visual-linguistic ambiguity, provide a small dataset containing 500 cooking videos with instruction and anaphora annotations. They also propose an end-to-end multitask learning framework for solving anaphora resolution as well as object localization.

**Reasons To Accept:**

1. The problem introduced is interesting and important. Formulating the ambiguities in the multi-modal space is important and this work takes a good step in this direction.
2. The dataset introduced will help researchers study the problem further and help the research take flight.
3. The approach proposed in the paper is also a solid baseline for this problem and shows good performance on the dataset.

**Reasons To Reject:**

1. Although the results show significant gains, the dataset is small and statistical significance is not provided. Thus, we cannot be sure if the approach would generalize to other datasets and if they are significant on this dataset.
2. Minor points
    1. In the abstract, the authors refer to the “visual-linguistic ambiguity” as chicken-and-egg problem. How is this chicken-and-egg? Oil mixed with salt and pepper becomes a mixture and there is a temporal relation among them.
    2. In line 227, the authors say the span might be an incorrect consecutive fragment. What do they mean by “incorrect” here?



**Reproducibility:**

4: Could mostly reproduce the results, but there may be some variation because of sample variance or minor variations in their interpretation of the protocol or method.

**Reviewer Confidence:**

4: Quite sure. I tried to check the important points carefully. It's unlikely, though conceivable, that I missed something that should affect my ratings.

---

> ### Author Rebuttal · Authors · 2023-08-28
>
> Many thanks for your careful and comprehensive feedback.
>
> ### Reasons To Reject #1:
>
> > More data is always better. One of the contributions of the paper is that we increase the original Chop&Change data set from 264 to 500 videos. However, annotations for video captions, anaphora resolution, and object localization are expensive. Even though 500 videos may still not sound that much, this provides us with 14,532 image-caption pairs (3 images from 4,633 caption-video pairs) to train our object localization model. To test the model we have 5,688 images. While statistical significance testing for object localization is difficult and we are not aware of any work yet that provides a statistical significance test for object localization, we will add statistical significance results for anaphora resolution obtained via the Approximate Randomization Test (usually used for coreference resolution as well). The $p$-value between the best and the second-best anaphora resolution results varies from 0.03 to 0.04 across the runs, hence our results are significant at the $p<0.05$ level.
>
> ### Reasons To Reject #2.1:
>
> > There is a temporal relation between **mix oil with salt and pepper** and **the mixture** and this temporal relation should be resolved to understand what **the mixture** is. But, we still cannot know which **mixture** is pointed out in the video. In image 1a in Figure 1, we resolve that **the pan** is the **the hot pan** thanks to temporal relations by using anaphora resolution. However, we cannot say which pan of the image (1b) is **the pan**. This causes the chicken-and-egg problem which describes the co-dependence between the two tasks and appears in the absence of a clear visual-linguistic alignment (please see lines 009-015).
>
> ### Reasons To Reject #2.2:
>
> > Let us assume we have a step like **add the onion pieces**. We face an incorrect consecutive fragment if the model detects **the onion** as a mention instead of  **the onion pieces**. This causes error propagation to the tasks of anaphora resolution and object localization.  Additionally, we agree that talking about the issue of incorrect spans at the beginning of the task description might be confusing for the readers. We will carry the part of incorrect spans from section 3. Task to section 5.1.1 Mention Extraction and Representation.

---

### Official Review · Reviewer_RhkM · 2023-08-05

**Soundness:** 3

**Excitement:**

4: Strong: This paper deepens the understanding of some phenomenon or lowers the barriers to an existing research direction.

**Paper Topic And Main Contributions:**

This paper focus on solving the vision and language ambiguities in multimodal learning tasks. It proposes a new task of multimodal resolution of vision and language ambiguities together with a dataset containing anaphora resolution and object localization tasks. This paper also proposes a new multitask learning system for modeling the two tasks jointly and shows that the two tasks can benefit each other.

**Reasons To Accept:**

1. The problem of vision and language ambiguities is important for multimodal learning research. The task and dataset proposed in this paper can benefit related research a lot.
2. This paper proposes a novel method to jointly model the anaphora resolution and object localization tasks. The experiments show the effectiveness of the proposed method. The authors also discuss the benefit of the tasks to each other, which can be a good reference for future research.
3. This paper is well-written and well-organized. The demonstration of the motivation and proposed dataset and methods are clear.

**Reasons To Reject:**

1. The experiment part should include more baseline methods to show the effectiveness of proposed method. It can implement some simple baselines based on existing vision-language models.
2. The experiment part should evaluate more vision-language models on the proposed task to give a more comprehensive analysis.

**Reproducibility:**

4: Could mostly reproduce the results, but there may be some variation because of sample variance or minor variations in their interpretation of the protocol or method.

**Reviewer Confidence:**

3: Pretty sure, but there's a chance I missed something. Although I have a good feel for this area in general, I did not carefully check the paper's details, e.g., the math, experimental design, or novelty.

---

> ### Author Rebuttal · Authors · 2023-08-28
>
> Many thanks for your feedback. We will provide a more comprehensive analysis of results.
>
> ### Reasons To Reject #1 and #2:
>
> > Because of the page limitation for the original submission, even though we had experimented with both FasterRCNN [2] and RegionCLIP [1], we had only included results with RegionCLIP as these showed state-of-the-art results for zero-shot object detection and open-vocabulary object detection [1]. We will use the additional page for a comprehensive analysis of results, including the experiments with FasterRCNN. We observed that RegionCLIP gives better region proposals compared to FasterRCNN.
>
> [1] Zhong, Y., Yang, J., Zhang, P., Li, C., Codella, N., Li, L. H., ... & Gao, J. (2022). Regionclip: Region-based language-image pretraining. In Proceedings of the IEEE/CVF Conference on Computer Vision and Pattern Recognition (pp. 16793-16803).
>
> [2] Ren, S., He, K., Girshick, R., & Sun, J. (2015). Faster r-cnn: Towards real-time object detection with region proposal networks. Advances in neural information processing systems, 28.

---

### Meta-Review · Area_Chair_ZSi5 · 2023-09-20

**Recommendation:** 5

**Metareview:**

This paper addresses an important yet previously overlooked issue of visual-linguistic ambiguity in multimodal understanding tasks. The authors have conducted thorough research, from task design, dataset construction, baseline method development, to empirical validation. This lays a solid foundation for subsequent research in the area. While there are some areas in the manuscript that could be improved, we look forward to an engaging camera-ready version that incorporates feedback from the reviews and discussion.

---

### Decision · Program_Chairs · 2023-10-07

**Decision:**

Accept-Main

**Comment:**

This paper addresses an important yet previously overlooked issue of visual-linguistic ambiguity in multimodal understanding tasks. The authors have conducted thorough research, from task design, dataset construction, baseline method development, to empirical validation. This lays a solid foundation for subsequent research in the area. While there are some areas in the manuscript that could be improved, we look forward to an engaging camera-ready version that incorporates feedback from the reviews and discussion.